# Risk Factors for Early Mortality in Older Patients with Traumatic Cervical Spine Injuries—A Multicenter Retrospective Study of 1512 Cases

**DOI:** 10.3390/jcm12020708

**Published:** 2023-01-16

**Authors:** Motoya Kobayashi, Noriaki Yokogawa, Satoshi Kato, Takeshi Sasagawa, Hiroyuki Tsuchiya, Hiroaki Nakashima, Naoki Segi, Sadayuki Ito, Toru Funayama, Fumihiko Eto, Akihiro Yamaji, Junichi Yamane, Satoshi Nori, Takeo Furuya, Atsushi Yunde, Hideaki Nakajima, Tomohiro Yamada, Tomohiko Hasegawa, Yoshinori Terashima, Ryosuke Hirota, Hidenori Suzuki, Yasuaki Imajo, Shota Ikegami, Masashi Uehara, Hitoshi Tonomura, Munehiro Sakata, Ko Hashimoto, Yoshito Onoda, Kenichi Kawaguchi, Yohei Haruta, Nobuyuki Suzuki, Kenji Kato, Hiroshi Uei, Hirokatsu Sawada, Kazuo Nakanishi, Kosuke Misaki, Hidetomi Terai, Koji Tamai, Akiyoshi Kuroda, Gen Inoue, Kenichiro Kakutani, Yuji Kakiuchi, Katsuhito Kiyasu, Hiroyuki Tominaga, Hiroto Tokumoto, Yoichi Iizuka, Eiji Takasawa, Koji Akeda, Norihiko Takegami, Haruki Funao, Yasushi Oshima, Takashi Kaito, Daisuke Sakai, Toshitaka Yoshii, Tetsuro Ohba, Bungo Otsuki, Shoji Seki, Masashi Miyazaki, Masayuki Ishihara, Seiji Okada, Shiro Imagama, Kota Watanabe

**Affiliations:** 1Department of Orthopaedic Surgery, Graduate School of Medical Sciences, Kanazawa University, Kanazawa 920-8641, Japan; 2Department of Orthopaedic Surgery, Toyama Prefectural Central Hospital, Toyama 930-8550, Japan; 3Department of Orthopedic Surgery, Graduate School of Medicine, Nagoya University, Nagoya 466-8550, Japan; 4Department of Orthopaedic Surgery, Faculty of Medicine, University of Tsukuba, Tsukuba 305-8575, Japan; 5Department of Orthopaedic Surgery, Graduate School of Comprehensive Human Sciences, University of Tsukuba, Tsukuba 305-8575, Japan; 6Department of Orthopaedic Surgery, Ibaraki Seinan Medical Center Hospital, Sakai 306-0433, Japan; 7Department of Orthopaedic Surgery, National Hospital Organization Murayama Medical Center, Tokyo 208-0011, Japan; 8Department of Orthopaedic Surgery, Keio University School of Medicine, Tokyo 160-8582, Japan; 9Department of Orthopaedic Surgery, Graduate School of Medicine, Chiba University, Chiba 260-8670, Japan; 10Department of Orthopaedics and Rehabilitation Medicine, Faculty of Medical Sciences, University of Fukui, Fukui 910-1193, Japan; 11Department of Orthopaedic Surgery, Hamamatsu University School of Medicine, Shizuoka 431-3192, Japan; 12Department of Orthopaedic Surgery, Nagoya Kyoritsu Hospital, Nagoya-shi 454-0933, Japan; 13Department of Orthopaedic Surgery, Sapporo Medical University, Sapporo 060-8543, Japan; 14Department of Orthopaedic Surgery, Matsuda Orthopedic Memorial Hospital, Sapporo 001-0018, Japan; 15Department of Orthopaedic Surgery, Yamaguchi University Graduate School of Medicine, Yamaguchi 755-8505, Japan; 16Department of Orthopaedic Surgery, Shinshu University School of Medicine, Nagano 390-8621, Japan; 17Department of Orthopaedics, Graduate School of Medical Science, Kyoto Prefectural University of Medicine, Kyoto 602-8566, Japan; 18Department of Orthopaedics, Saiseikai Shiga Hospital, Ritto 520-3046, Japan; 19Department of Orthopaedic Surgery, Tohoku University Graduate School of Medicine, Sendai 980-8574, Japan; 20Department of Orthopaedic Surgery, Graduate School of Medical Sciences, Kyushu University, 3-1-1 Maidashi Higashi-ku, Fukuoka 812-8582, Japan; 21Department of Orthopaedic Surgery, Nagoya City University Graduate School of Medical Sciences, Nagoya 467-8601, Japan; 22Department of Orthopaedic Surgery, Nihon University Hospital, Tokyo 101-8393, Japan; 23Department of Orthopaedic Surgery, Nihon University School of Medicine, Tokyo 173-8610, Japan; 24Department of Orthopedics, Traumatology and Spine Surgery, Kawasaki Medical School, Okayama 701-0192, Japan; 25Department of Orthopaedic Surgery, Osaka Metropolitan University Graduate School of Medicine, Osaka 545-8585, Japan; 26Department of Orthopaedic Surgery, Kitasato University School of Medicine, Sagamihara 252-0374, Japan; 27Department of Orthopaedic Surgery, Kobe University Graduate School of Medicine, Kobe 650-0017, Japan; 28Department of Orthopaedic Surgery, Kochi Medical School, Kochi University, Nankoku 783-8505, Japan; 29Department of Orthopaedic Surgery, Graduate School of Medical and Dental Sciences, Kagoshima University, Kagoshima 890-8520, Japan; 30Department of Orthopaedic Surgery, Gunma University Graduate School of Medicine, Maebashi 371-8511, Japan; 31Department of Orthopaedic Surgery, Mie University Graduate School of Medicine, Mie Tsu City 514-8507, Japan; 32Department of Orthopaedic Surgery, School of Medicine, International University of Health and Welfare, Chiba 286-0124, Japan; 33Department of Orthopaedic Surgery, International University of Health and Welfare Narita Hospital, Chiba 286-0124, Japan; 34Department of Orthopaedic Surgery and Spine and Spinal Cord Center, International University of Health and Welfare Mita Hospital, Tokyo 108-8329, Japan; 35Department of Orthopaedic Surgery, The University of Tokyo Hospital, Tokyo 113-8655, Japan; 36Department of Orthopaedic Surgery, Osaka University Graduate School of Medicine, Osaka 565-0871, Japan; 37Department of Orthopedics Surgery, Surgical Science, Tokai University School of Medicine, Isehara 259-1193, Japan; 38Department of Orthopaedic Surgery, Tokyo Medical and Dental University, Tokyo 113-8519, Japan; 39Department of Orthopaedic Surgery, University of Yamanashi, Yamanashi 409-3898, Japan; 40Department of Orthopaedic Surgery, Graduate School of Medicine, Kyoto University, Kyoto 606-8507, Japan; 41Department of Orthopaedic Surgery, Faculty of Medicine, University of Toyama, Toyama 930-0194, Japan; 42Department of Orthopaedic Surgery, Faculty of Medicine, Oita University, Yufu-shi 879-5593, Japan; 43Department of Orthopaedic Surgery, Kansai Medical University Hospital, Osaka 573-1191, Japan

**Keywords:** risk factor, cervical spine injury, chronic kidney disease, prognostic factor, early mortality

## Abstract

For older patients with decreased reserve function, traumatic cervical spine injuries frequently lead to early mortality. However, the prognostic factors for early mortality remain unclear. This study included patients aged ≥65 years and hospitalized for treatment of traumatic cervical spine injuries in 78 hospitals between 2010 and 2020. Early mortality was defined as death within 90 days after injury. We evaluated the relationship between early mortality and the following factors: age, sex, body mass index, history of drinking and smoking, injury mechanisms, presence of a cervical spine fracture and dislocation, cervical ossification of the posterior longitudinal ligament, diffuse idiopathic skeletal hyperostosis, American Spinal Injury Association Impairment Scale, concomitant injury, pre-existing comorbidities, steroid administration, and treatment plan. Overall, 1512 patients (mean age, 75.8 ± 6.9 years) were included in the study. The early mortality rate was 4.0%. Multivariate analysis identified older age (OR = 1.1, *p* < 0.001), male sex (OR = 3.7, *p* = 0.009), cervical spine fracture (OR = 4.2, *p* < 0.001), complete motor paralysis (OR = 8.4, *p* < 0.001), and chronic kidney disease (OR = 5.3, *p* < 0.001) as risk factors for early mortality. Older age, male sex, cervical spine fracture, complete motor paralysis, and chronic kidney disease are prognostic factors for early mortality in older patients with traumatic cervical spine injuries.

## 1. Introduction

Population aging is a global phenomenon. Epidemiological studies show that the percentage of the population aged over 65 years increased from 6% in 1990 to 9% in 2019. This figure is projected to rise further to 16% by 2050 [1]. Furthermore, with the increased average life expectancy, older individuals are more active today than in past generations, putting them at greater risk of traumatic musculoskeletal injuries [2].

Traumatic cervical spine injuries are serious injuries caused by even low-energy falls from standing height [3,4], representing a significant cause of morbidity and mortality among older adults [5,6]. Therefore, in Japan, where population aging is progressing more rapidly than in other countries, managing traumatic cervical spine injuries in older patients is becoming an important topic of debate.

Traumatic cervical spine injuries frequently accompany spinal cord injuries, which cause respiratory complications secondary to the disruption of innervation to the respiratory muscles [7,8]. Furthermore, spinal cord injury causes cardiovascular system complications due to the loss of supra-spinal control of the autonomic nerves [9,10]. Moreover, surgical treatment to provide stability is frequently required in patients with cervical spine fractures, increasing the risk of perioperative complications due to surgical invasiveness [11,12]. Even if conservative treatment is chosen, complications correlated with long-term bed rest and cervical orthosis are a concern [13,14]. In older patients with decreased reserve function and additional medical comorbidities, these complications associated with traumatic cervical spine injuries contribute to early mortality. Previous studies revealed that an early mortality rate after traumatic cervical spine injuries in older patients was significantly higher than that in younger patients [15,16,17,18,19,20].

However, predictors of early mortality in older patients with traumatic cervical spine injuries have not been accurately identified. A better understanding of the factors related to early mortality may help physicians make effective management decisions. Therefore, this study aimed to determine the predictors of early mortality after traumatic cervical spine injuries in older individuals.

## 2. Materials and Methods

### 2.1. Study Design

Patients aged ≥65 years who received inpatient treatment for traumatic cervical spine injuries at 78 institutions between 2010 and 2020 and were followed for at least 3 months were included in this study. Patients whose treatment course and outcome were unclear were excluded. Data collection was performed using the multicenter database of the Japan Association of Spine Surgeons with Ambition [21]. Traumatic cervical spine injuries included (1) isolated cervical fractures and/or dislocations, (2) isolated spinal cord injury, and (3) spinal cord injury with fractures and/or dislocations. The Institutional Review Board of Kanazawa University, the representative institution, approved this study protocol (No. 3352-1). 

### 2.2. Variables and Outcomes

Clinical variables included age at injury, sex, body mass index, history of smoking and drinking, injury mechanism, presence of cervical spine fracture and dislocation, presence of cervical ossification of the posterior longitudinal ligament (OPLL), diffuse idiopathic skeletal hyperostosis (DISH), American Spinal Injury Association Impairment Scale (AIS) score at admission, presence of concomitant injury, pre-existing comorbidities, presence of steroid administration, and treatment plan (surgical and conservative treatment). Pre-existing comorbidities included cerebrovascular disease, respiratory disease, diabetes mellitus, hypertension, cardiac disease, and chronic kidney disease. Surgical-related variables included the presence of early surgery intervention, surgical method (single-side and combined approach), presence of instrumentation, operative time, intraoperative blood loss, presence of perioperative blood transfusion, and intraoperative complications. Early surgery intervention was defined as surgery within 24 h after injury. Intraoperative complications included massive bleeding (>1000 g) [22], dural tear, intraoperative vertebral artery injury, intraoperative injury of the spinal cord or nerve root, and intraoperative spine fracture. The outcome of interest was early mortality, which was defined as death within 90 days after injury.

### 2.3. Statistical Analysis

The patients were categorized into survival and early-mortality groups. We compared clinical variables between the two groups. Mean and standard deviations, for continuous variables, or number (percentage), for categorical variables are reported. The normality of the distribution of continuous data was assessed using the Shapiro–Wilk test. Differences in continuous variables were examined using the Student *t*-test and Mann–Whitney U for parametric and nonparametric data, respectively. The chi-square or Fisher’s exact test was used to analyze categorical data. All *p*-values were 2-tailed, and the statistical significance was set at *p* < 0.05. Multivariate analysis used a multivariable stepwise logistic regression to identify independent risk factors for early mortality. The variables with univariate *p* < 0.05 were candidates for multiple logistic regression. A pairwise deletion was performed in the univariate analysis to address missing data. In multiple logistic regression analysis, the listwise deletion of missing data was performed. IBM SPSS Statistics for Windows, version 25 (IBM Corp., Armonk, NY, USA) was used for statistical analysis. Moreover, we performed a subgroup analysis of patients who underwent surgical treatment for traumatic cervical spine injuries. The subgroup analysis aimed to explore surgery-related variables associated with early mortality. This study required at least 1434 cases to achieve a medium effect (effect size, d  =  0.25), a power of 0.90, and a statistical significance level of 0.05, according to the findings of a nationwide study that early mortality after traumatic cervical spine injuries in older patients was 11.3% [18]. In sample size calculation, we used the G-power software (Franz Faul, Univesitat kiel, Germany, Kiel).

## 3. Results

### 3.1. Overall Outcome

Overall, 1512 patients (1007 males and 505 females; mean age, 75.8 ± 6.9 years; mean follow-up, 19.1 ± 21 months) were included in this study. One thousand and forty-seven patients sustained neurological injuries: 127 complete and 920 incomplete. Cervical spine fractures and dislocations were observed in 834 and 228 cases, respectively. The most common injury mechanism was falling on level ground (579 cases), followed by falls from 1 m or more (333 cases), traffic accidents (287 cases), falls from 1 m or less (241 cases), and others (72 cases). Cervical OPLL and DISH were found in 332 and 196 cases, respectively. The most common pre-existing comorbidities were hypertension (731 cases), followed by diabetes mellitus (330 cases), cardiac disease (227 cases), cerebrovascular disease (145 cases), respiratory disease (81 cases), and chronic kidney disease (46 cases). Additionally, 903 and 609 patients received surgical and conservative treatment, respectively.

Early mortality was observed in 61 cases (4.0%). The mortality rates were 0.6% (2/321), 3.2% (12/376), 3.1% (11/356), 5.4% (15/280), 9.2% (12/130), and 18.4% (9/49) in the 65–69, 70–74, 75–79, 80–84, 85–89, and >90 years age groups, respectively. The causes of early mortality are presented in Table 1. The most common deaths were respiratory-related deaths (24 cases, 39.3%), followed by cardiovascular- (11 cases, 18.0%) and gastrointestinal-related deaths (4 cases, 6.6%). 

The results of the comparison between the survival and early mortality groups are presented in Table 2. Older age (*p* < 0.001), male sex (*p* = 0.001), presence of cervical spine fracture (*p* < 0.001), presence of DISH (*p* = 0.01), complete motor paralysis (AIS A or B) (*p* < 0.001), respiratory disease (*p* = 0.03), cardiac disease (*p* < 0.001), chronic kidney disease (*p* = 0.001), and conservative treatment (*p* = 0.001) are associated with early mortality. Multivariate regression analysis was performed on the factors that were significant in the univariate analysis. Older age (odds ratio (OR) = 1.1, 95% confidence interval (CI) = 1.1–1.2, *p* < 0.001), male sex (OR = 3.7, 95% CI = 1.5–9.3, *p* = 0.009), presence of cervical spine fracture (OR = 4.2, 95% CI = 2.0–9.0, *p* < 0.001), complete motor paralysis (AIS A or B) (OR = 8.4, 95% CI = 4.3–16.3, *p* < 0.001), and chronic kidney disease (OR = 5.3, 95% CI = 2.1–14.0, *p* < 0.001) were identified as independent risk factors for early mortality after traumatic spine injury in older patients. 

### 3.2. Sub-Group Analysis

A total of 903 patients who underwent surgery (631 males and 272 females; mean age, 74.9 ± 6.3 years; mean follow-up, 20.0 ± 20 months) were included in the subgroup analysis. In addition, 637 patients sustained neurological injuries: 88 complete and 549 incomplete. Cervical spine fractures and dislocations were observed in 531 and 211 cases, respectively. The operative treatment consisted of 66 anterior, 811 posterior, and 23 combined procedures (three cases were unknown). Instrumentation was performed in 628 cases. The mean operative time was 168 ± 74 min. The mean intraoperative bleeding volume was 227 ± 353 mL, and 106 patients required blood transfusion perioperatively. Intraoperative complications were found in 44 patients (Table 3).

Early mortality was observed in 24 cases (2.7%). The most common causes of death were respiratory-related (nine cases, 37.5%), including six cases of pneumonia, two respiratory failures, and one case of choking. The second most common causes of death were cardiovascular-related (eight cases, 33.3%), including three strokes, three heart failures, one myocardial infarction, and one pulmonary embolism. The other causes of death were gastrointestinal bleeding (two deaths), non-occlusive mesenteric ischemia (one death), hemorrhagic shock (one death), and unknown (three deaths).

The results of the comparison between the survival and early mortality groups are shown in Table 4. Older age (*p* = 0.01), the presence of DISH (*p* = 0.04), complete motor paralysis (AIS A or B) (*p* < 0.001), diabetes mellitus (*p* = 0.04), cardiac disease (*p* = 0.002), perioperative blood transfusion (*p* < 0.001), and intraoperative complications (*p* = 0.03) were associated with early mortality. In multivariate regression analysis, older age (OR = 1.1, 95% CI = 1.0–1.2, *p* = 0.01), complete motor paralysis (AIS A or B) (OR = 6.8, 95% CI = 2.7–17.7, *p* < 0.001), cardiac disease (OR = 5.2, 95% CI = 2.0–13.5, *p* < 0.001), and intraoperative complications (OR = 5.1, 95% CI = 1.5–17.7, *p* = 0.01) were identified as independent risk factors for early mortality after traumatic spine injury in older patients. 

## 4. Discussion

In treating traumatic cervical spine injuries in older patients, a better understanding of prognostic factors for early mortality plays a vital role in devising an appropriate treatment strategy for each individual and preventing early mortality. Recently, several studies have attempted to identify predictors of early mortality in older patients with traumatic cervical spine injuries. Several factors, including older age, level of injury, the severity of spinal cord injury, sex, Glasgow Coma Scale score, impaired hemodynamics at admission, application of respiratory assistance, and pre-existing comorbidities are related to early mortality after traumatic cervical spine injuries in older patients [15,16,17,19,20]. However, few studies had significant statistical power and controlled for potential confounders. Asemota et al. retrospectively investigated in-hospital mortality after traumatic cervical spine injuries in older patients using a nationwide database of 167,278 older adults. However, their study was limited to an epidemiological investigation without a multivariate analysis for exploring prognostic factors for early mortality [18]. Therefore, to the best of our knowledge, this is the largest (*n* = 1512 older patients) retrospective multicenter study to investigate the risk factors for early mortality after traumatic cervical spine injuries in older patients.

This study revealed that complete motor paralysis with AIS scores of A and B resulted in an 8.4-fold-higher risk of early mortality. Generally, spinal cord injury causes respiratory muscle dysfunction and the loss of supra-spinal control of the autonomic nerves, resulting in complications related to the respiratory and cardiovascular systems [7,8,9,10]. It is reasonable that the severity of spinal cord injuries was strongly related to early mortality since these complications are the leading causes of death in the acute phase of traumatic cervical spine injuries [23,24]. Similarly, a previous retrospective study including patients aged >60 years with traumatic cervical spine injuries reported that complete spinal cord injury was associated with a 5.1-times-higher risk of in-hospital mortality than incomplete spinal cord injury [17].

Numerous previous studies reported that pre-existing comorbidities were associated with early mortality after traumatic cervical spine injuries in older patients [20,25,26]. However, details of pre-existing comorbidities influencing prognosis are yet to be identified. This is the first study to explore the types of pre-existing comorbidities associated with early mortality after traumatic spine injury in older patients, which identified chronic kidney disease as an independent risk factor for early mortality in the multivariate analysis. 

Consistent with this finding, previous studies in the field of spine surgery also found a negative prognostic impact of chronic kidney disease. For example, a multicenter retrospective study including 26,604 patients demonstrated that preoperative renal comorbidity was an independent risk factor for in-hospital mortality after spinal surgery (OR = 4.3, 95% CI = 1.4–13.6) [27]. Similarly, a retrospective nationwide cohort study showed that spinal cord injury with chronic kidney disease had a significantly higher 1-year mortality rate than spinal cord injury without chronic kidney disease (17.7% vs. 8.5%) [28].

This may be because chronic kidney disease is associated with other pre-existing comorbidities. Patients with chronic kidney disease have comorbidities that could contribute to or be caused by their kidney disease. Han et al. retrospectively reviewed patients with chronic renal failure who underwent elective spinal surgery, demonstrating that the cohort had a high rate of multiple medical comorbidities, including hypertension, diabetes mellitus, vascular disease, preoperative anemia, and hyperkalemia [29]. Furthermore, in this study, the cohort with chronic kidney disease had an average of 1.78 additional comorbidities other than chronic kidney disease. The multiple pre-existing comorbidities related to chronic renal disease may be associated with the development of perioperative complications and a poor prognosis.

We found that the presence of cervical spine fracture considerably impacted the risk of early mortality after traumatic cervical spine injuries. Cervical fractures frequently require surgery to provide stability to the injured spine, which increases the risk of perioperative complications due to the invasiveness of surgery [11,12]. Even if conservative treatment is chosen, long-term bed rest with a hard cervical collar or halo-vest fixation leads to various complications, including pneumonia, venous thromboembolism, and pressure ulcers [13,14]. In addition, recent studies have found a risk of acute mortality of more than 20% for older patients with isolated cervical spine fractures [30].

Interestingly, the male sex was also a significant risk factor associated with poor prognosis after traumatic cervical spine injuries. Similarly, a retrospective analysis of 1995 adults with traumatic cervical spine injuries showed a relationship between the male sex and increased early mortality [31]. Furthermore, in traumatic brain injury, previous research reported that the female sex was independently associated with reduced mortality and decreased complications [32,33].

In this study, older age is also a causative factor related to poor prognosis in traumatic cervical spine injuries, which is consistent with the findings of previous studies [18,20,34,35]. Various factors derived from advanced age, including osteoporotic bones, reduced recovery capacity, and medical comorbidities, may be associated with early mortality [34]. Martin et al. investigated early mortality rates after traumatic cervical spine injuries in different age groups. They demonstrated that the increase in the mortality curve becomes statistically significant from the age of 70 years [35]. A similar trend was observed in this study, with early mortality rates of 0.6% (2/321) and 5.0% (59/1191) for the 65–69 and ≥70 years age groups, respectively.

The early mortality rate in this study was relatively low at 4.0%, whereas the reported early mortality rate after traumatic cervical spine injuries among older patients is 9.6–38.0% [15,16,17,18,19,20]. This may be because the cohort in this study was younger and had lower rates of complete spinal cord injury and cervical spine fracture than those in previous studies (Table 5). Another possible reason was that the cohort was relatively recent, including cases from 2010 to 2020. Previous studies reported a decreased mortality rate for traumatic cervical spine injuries during recent periods due to advancing medical management [36,37]. Moreover, missing data due to the design of the retrospective multicenter study may have led to a low estimate of the occurrence of early mortality.

Although many cases of traumatic cervical spine injuries require surgical treatment, few reports have focused on the impact of surgery-related factors on early mortality in older patients with traumatic cervical spine injuries. Therefore, we performed a subgroup analysis limited to patients who underwent surgical treatment to analyze the association between early mortality and clinical variables, including surgery-related factors.

Among the surgery-related variables, intraoperative complications were the only risk factor for early mortality. In this study, 44 intraoperative complications occurred, of which four cases resulted in early mortality. Notably, in two of the four early-mortality cases, intraoperative complications were directly related to the cause of death. One patient with massive intraoperative bleeding died intraoperatively due to hemorrhagic shock, and the other with an intraoperative vertebral artery injury died 5 days postoperatively due to brainstem infarction (Table 3). These two cases show that intraoperative complications can directly and significantly impact early mortality in older patients with traumatic cervical spine injuries. In the surgical treatment of traumatic cervical spine injuries, various factors, including traumatic anatomical changes, instrumentation procedures for traumatic spinal instability, and insufficient withdrawal period of antiplatelet or anticoagulant medications in emergency surgery, increase the risk of intraoperative complications [38,39,40]. Therefore, the surgeon should take measures, including appropriate surgical approaches and accurate preoperative imaging evaluation, to avoid intraoperative complications.

Cardiac disease was also identified as an independent risk factor for early mortality in the surgical treatment of traumatic cervical spine injuries among older patients. Generally, surgical treatment accompanies blood loss and perioperative blood pressure fluctuations, which significantly burden the cardiac system. Older patients with cardiac disease have a lower tolerance for this hemodynamic stress, which may increase the risk of early mortality. A similar association between cardiac disease and increased early mortality after surgical treatment was found in hip fracture treatment [41,42]. In addition, spine surgery is typically longer than other elective orthopedic surgeries and is frequently accompanied by significant blood loss, fluid shifts, and transfusion [43]. Therefore, it is reasonable that cardiac disease was associated with early mortality after surgical treatment for traumatic cervical spine injuries in older patients. Supporting this hypothesis, a retrospective study using a nationwide database demonstrated that comorbid congestive heart failure was associated with increased in-hospital mortality after spine surgery (OR = 4.6, 95% CI = 3.8–5.5) [44].

This study has some limitations. Firstly, due to the nature of retrospective multicenter studies, the number of excluded patients and their baseline characteristics are unknown, and there may be some missing data. In addition, treatment strategies may differ between institutions. Secondly, the selection criteria for this study excluded deaths before hospital transport or in the emergency room, which underestimates early mortality. Thirdly, there is a bias regarding surgical indications. The fact that surgical treatment was associated with lower early mortality in the univariate analysis may be attributed to selection bias, where patients who could tolerate surgical intervention were selected. Therefore, these data should be carefully interpreted. Fourthly, the number of early mortalities was small. Therefore, further studies with larger sample sizes are needed to ensure the generalizability of our conclusions. Despite these limitations, this study presents informative data that are useful in clinical practice, particularly for spine surgeons who manage traumatic cervical spine injuries in older patients.

## 5. Conclusions

Complete motor paralysis, chronic kidney disease, the presence of cervical spine fracture, male sex, and older age were identified as risk factors for early mortality after traumatic cervical injuries in older patients. In addition to complete motor paralysis and older age, intraoperative complications and cardiac disease were identified as independent risk factors for early mortality in the surgical treatment of traumatic cervical spine injuries among older patients.

## Figures and Tables

**Table 1 jcm-12-00708-t001:** Causes of early mortality after traumatic cervical spine injury in older patients.

Cause of Death	Number
Respiratory	24
- Pneumonia;	14
- Respiratory failure;	6
- Choke.	4
Cardiovascular	11
- Stroke;	6
- Cardiac failure;	3
- Myocardial infarction;	1
Pulmonary embolism.	1
- Gastrointestinal	4
- Gastrointestinal bleeding;	3
- Non-occlusive mesenteric ischemia.	1
Systemic	7
- Hemorrhagic shock;	3
- Renal failure;	2
- Spinal shock;	1
- Urinary tract infection.	1
Unknown	15

**Table 2 jcm-12-00708-t002:** Comparison between the survival and early mortality groups.

Clinical Variables	Early Mortality (*n* = 61)	Survival (*n* = 1451)	*p*
Age at injury (mean ± SD)	80.9 ± 7.3	75.6 ± 6.9	<0.001 *
Sex			0.001 *
- Male (%)	85.2% (52/61)	65.8% (955/1451)	
- Female (%)	14.8% (9/61)	34.2% (496/1451)	
BMI (mean ± SD)	21.2 ± 4.2	22.1 ± 3.6	0.08
Smoking history (%)	31.4% (11/35)	29.9% (277/927)	0.85
Drinking history (%)	27.2% (9/33)	38.2% (339/887)	0.27
Injury mechanism			
- Falling from the level ground (%)	39.3% (24/61)	38.4% (555/1444)	0.89
- Others (%)	60.7% (37/61)	61.6% (889/1444)	
Cervical spine fracture (%)	80.3% (49/61)	54.1% (785/1451)	<0.001 *
Cervical spine dislocation (%)	21.3% (13/61)	14.8% (215/1451)	0.20
Cervical OPLL (%)	23.0% (14/61)	21.9% (318/1450)	0.88
DISH (%)	26.8% (15/56)	13.8% (181/1308)	0.01 *
AIS at the admission			<0.001 *
- A or B (%)	42.4% (25/59)	12.3% (178/1445)	
- C, D, or E (%)	57.6% (34/59)	87.7% (1267/1445)	
Concomitant injury (%)	37.7% (23/61)	26.7% (388/1451)	0.08
Cerebrovascular disease (%)	14.0% (8/57)	9.8% (137/1402)	0.26
Respiratory disease (%)	12.5% (7/56)	5.3% (74/1399)	0.03 *
Diabetes mellitus (%)	28.1% (16/57)	22.4% (314/1401)	0.33
Hypertension (%)	54.4% (31/57)	49.4% (700/1417)	0.50
Cardiac disease (%)	33.3% (19/57)	14.9% (208/1399)	<0.001 *
Chronic kidney disease (%)	12.5% (7/56)	2.8% (39/1398)	0.001 *
Steroid administration (%)	11.5% (7/61)	13.8% (200/1450)	0.71
Surgical treatment (%)	39.3% (24/61)	60.6% (879/1451)	0.001 *

SD, standard deviation; BMI, body mass index; OPLL, ossification of the posterior longitudinal ligament; DISH, diffuse idiopathic skeletal hyperostosis; AIS, American Spinal Injury Association Impairment Scale; * *p* < 0.05.

**Table 3 jcm-12-00708-t003:** Details of intraoperative complications.

Intraoperative Complications	Number	Note
Massive bleeding (>1000 g)	18	One patient died intraoperatively due to hemorrhagic shock
Dural tear	16	
Intraoperative vertebral artery injury	5	One patient died five days postoperatively due to brainstem infarction
Intraoperative injury of the spinal cord or nerve root	3	
Intraoperative spine fracture	2	

**Table 4 jcm-12-00708-t004:** Comparison between the survival and early mortality groups in the surgical treatment group.

	Early Mortality (*n* = 24)	Survival (*n* = 879)	*p*
Age at injury (mean ± SD, year)	78.4 ± 6.5	74.9 ± 6.3	0.01 *
Sex			0.18
- Male (%)	83.3% (20/24)	69.5% (611/879)	
- Female (%)	16.7% (4/24)	30.5% (268/879)	
BMI (mean ± SD)	22.4 ± 4.1	22.1 ± 3.4	0.99
Smoking history (%)	35.7% (5/14)	31.6% (182/575)	0.78
Drinking history (%)	16.7% (2/12)	40.0% (223/557)	0.14
Injury mechanism			0.83
- Falling from the level ground (%)	33.3% (8/24)	37.3% (326/873)	
- Others (%)	66.7% (16/24)	62.7% (547/873)	
Cervical spine fracture (%)	79.2% (19/24)	58.2% (512/879)	0.06
Cervical spine dislocation (%)	33.3% (8/24)	23.1% (203/879)	0.23
Cervical OPLL (%)	33.3% (8/24)	24.7% (217/878)	0.34
DISH (%)	36.4% (8/22)	17.4% (136/779)	0.04 *
AIS at the admission			<0.001 *
- A or B (%)	54.2% (13/24)	14.6% (128/876)	
- C, D, or E (%)	45.8% (11/24)	85.4% (748/876)	
Concomitant injury (%)	20.8% (5/24)	24.1% (212/879)	0.81
Cerebrovascular disease (%)	4.3% (1/23)	5.2% (44/847)	1.00
Respiratory disease (%)	20.8% (5/24)	9.4% (80/850)	0.08
Diabetes mellitus (%)	42.7% (10/24)	22.1% (187/848)	0.04 *
Hypertension (%)	54.2% (13/24)	48.7% (419/860)	0.68
Cardiac disease (%)	41.7% (10/24)	15.2% (129/847)	0.002 *
Chronic kidney disease (%)	8.3% (2/24)	2.5% (21/846)	0.13
Steroid administration (%)	8.3% (2/24)	11.8% (104/879)	1.00
Early surgery intervention (%)	13.0% (3/23)	9.7% (84/865)	0.49
Surgical method			0.47
- Single-side (%)	95.8% (23/24)	97.5% (854/876)	
- Combined (%)	4.2% (1/24)	2.5% (22/876)	
Instrumentation (%)	87.5 % (21/24)	69.3% (607/876)	0.07
Operative time (mean ± SD, min)	191 ± 91	167 ± 73	0.19
Intraoperative blood loss (mean ± SD, mL)	553 ± 962	218 ± 319	0.07
Perioperative blood transfusion (%)	39.1% (9/23)	11.4% (97/848)	<0.001 *
Intraoperative complications (%)	16.7% (4/24)	4.6% (40/866)	0.03 *

SD, standard deviation; BMI, body mass index; OPLL, ossification of the posterior longitudinal ligament; DISH, diffuse idiopathic skeletal hyperostosis; AIS, American Spinal Injury Association Impairment Scale; * *p* < 0.05.

**Table 5 jcm-12-00708-t005:** Previous research focusing on early mortality after traumatic cervical spine injury in older patients.

Study	Study Design	Number	Early Mortality Rate	Definition of Early Mortality	Average Age	Sex (Male)	Cervical Spine Fracture	Complete Spinal Cord Injury	Incomplete Spinal Cord Injury	Neurologically Intact
Jacskon et al., 2005 [15]	Retrospective single-center study	74	12.2%	Death in hospital	N/A	N/A	N/A	27.0%	40.5%	32.4%
Sokolowski et al., 2007 [16]	Retrospective single-center study	193	14.0%	Death in hospital	N/A	N/A	N/A	17.1%	35.6%	47.2%
Daneshvar et al., 2013 [17]	Two-center retrospective study	37	38.0%	Death in hospital	75.0	78.4%	100%	40.5%	59.5%	0%
Asemota et al., 2018 [18]	Retrospective study using a nationwide database	167,278	11.3%	Death in hospital	81.0	75.4%	100%	0.9%	7.8%	91.3%
Bokhari et al., 2018 [19]	Retrospective single-center study	225	14.2%	Death in hospital	79.7	50.2%	97.3%	2.2%	13.8%	84.0%
Inglis et al., 2020 [20]	Retrospective multicenter study	1340	9.6%	Death in hospital	74.6	71.4%	56.8%	17.5%	61.2%	21.3%
This study	Retrospective multicenter study	1512	4.0%	Death within 90 days after injury	75.8	66.6%	55.2%	8.4%	60.9%	30.7%

N/A, not applicable.

## Data Availability

The datasets used and analyzed during the current study are available from the corresponding author on reasonable request.

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
