# Peer review of "Risk Factors for Early Mortality in Older Patients with Traumatic Cervical Spine Injuries—A Multicenter Retrospective Study of 1512 Cases"

_jcm, 2023, doi:10.3390/jcm12020708_

Round 1

Reviewer 1 Report

Dear Editor,

I read the manuscript titled “Risk factors for early mortality in older patients with traumatic cervical spine injuries -A multicenter retrospective study of 1,512 cases-”. The authors conducted a retrospective case-control study based on a multicenter study population to identify potential mortality modifiers after cervical spine trauma. The authors reported that advanced age, male gender, cardiac disease and DM, neurological deficits after injury, and intraoperative transfusion requirement were all associated with increased early mortality.

The strengths of the current draft include the following:

1.     The topic is of significant clinical relevance, and the findings are expected to influence future medical practice

2.     The conclusions are justified by the available data and analysis

3.     The study sample is large from database evidence of multiple centres

4.     A study design is relatively strong as it is an Observational study (case-control)

5.     The authors performed all necessary subgroup analysis

6.     The manuscript is Well-written

7.     The manuscript is Supported by the appropriate statistics, tables, and references

8.     The were no ethical issues concerning the present study

At the same time, the study has some inherent limitations

1.     Potential introduction of bias due to the absence of comparability of the two arms on baseline characteristics (not nonrandomized)

2.     Potential introduction of bias due to the heterogeneity in the management (different centres and treatment policies)

The authors are advised to:

1. Describe the number of patients excluded from the study, the reason for exclusion, and their baseline characteristics, if possible.  

2.     Explain how they treated cases with missing data.

The recommended changes are not anticipated to influence the content and structure of the manuscript significantly. Therefore, I recommend a minor revision before reconsidering the publication of the manuscript.

Best regards

Author Response

Dear Editor,

We would like to thank you and the reviewers for all your time and effort devoted to reviewing our manuscript titled ‘‘Risk factors for early mortality in older patients with traumatic cervical spine injuries -a multicenter retrospective study of 1,512 cases.’’ The reviewers’ comments were indeed insightful and greatly appreciated. Therefore, we would like to take this opportunity to address every concern that the reviewers noted in their review of our submission. In addition, where appropriate, we have revised our manuscript accordingly. Any changes in this paper are marked. We believe that the reviewers’ comments have helped improve the quality of our manuscript. We hope that you and the reviewers will find our revised work suitable for publication in Journal of Clinical Medicine.

Thank you for your insightful comments. Our detailed responses and comments are presented below. (We have attached the revised manuscript for your reference.)

Response to Reviewer 1

# Comment 1

Describe the number of patients excluded from the study, the reason for exclusion, and their baseline characteristics, if possible.  

Response

We thank you for your insightful comment. In this study, we excluded patients whose course of treatment and outcome were unclear, even if they met the inclusion criteria. However, the number of excluded patients and their baseline characteristics were unknown because of the nature of the retrospective multicenter study. We consider this a limitation of this study.

To improve the manuscript, we have revised the sentences.

Patients whose treatment course and outcome were unclear were excluded. (P3 L155–156, blue font)

This study had some limitations. First, due to the nature of retrospective multicenter studies, the number of excluded patients and their baseline characteristics were unknown, and there may be some missing data. (P10 L409–410, blue font)

#Comment 2

Explain how they treated cases with missing data.

Response

We thank you for your insightful comment. A pairwise deletion was performed in the univariate analysis to address missing data. In multiple logistic regression analysis, listwise deletion of missing data was performed.

To improve the manuscript, we have modified the sentences. (P4 L191–192, blue font)

A pairwise deletion was performed in univariate analysis to address missing data. In multiple logistic regression analysis, listwise deletion of missing data was performed.

We appreciate your reconsideration of our manuscript. We believe that this study’s topic will interest your readers and that the data are informative. We hope that the revised manuscript is suitable for publication in your journal.

Sincerely,

Reviewer 2 Report

This multicenter retrospective study aims to determine the predictors of early mortality after traumatic cervical spine injuries in older persons. The conclusion of this study was Complete motor paralysis, chronic kidney disease, presence of cervical spine fracture, male sex, and older age were identified as risk factors for early mortality after traumatic cervical injuries in older patients. In addition to complete motor paralysis and older age, intraoperative complications and cardiac disease were identified as independent risk factors for early mortality in the surgical treatment of traumatic cervical spine injuries among older patients. This study involved relatively large sample sizes from multicenter data. There exist some shortcomings as detailed in the following.

1. Please state the method of sample size calculation in this study.
2. A description of the missing data needs to be mentioned.
3. The discussion section is too large, please reduce it. For example, there is no need to repeat the result part again.
4. Language editing and spell check are required.

Author Response

Dear Editor,

We would like to thank you and the reviewers for all your time and effort devoted to reviewing our manuscript titled ‘‘Risk factors for early mortality in older patients with traumatic cervical spine injuries -a multicenter retrospective study of 1,512 cases.’’ The reviewers’ comments were indeed insightful and greatly appreciated. Therefore, we would like to take this opportunity to address every concern that the reviewers noted in their review of our submission. In addition, where appropriate, we have revised our manuscript accordingly. Any changes in this paper are marked. We believe that the reviewers’ comments have helped improve the quality of our manuscript. We hope that you and the reviewers will find our revised work suitable for publication in Journal of Clinical Medicine.

Thank you for your insightful comments. Our detailed responses and comments are presented below. (We have attached the revised manuscript for your reference.)

Response to Reviewer 2
# Comment 1

Please state the method of sample size calculation in this study.

Response

We thank you for your valuable comment. This study required at least 1434 cases to achieve a medium effect (effect size, d = 0.25), a power of 0.90, and a statistical significance level of 0.05, considering the finding of a nationwide study that early mortality after traumatic cervical spine injuries in older patients was 11.3% [1].

  1. Asemota, A.O.; Ahmed, A.K.; Purvis, T.E.; Passias, P.G.; Goodwin, C.R.; Sciubba, D.M. Analysis of cervical spine injuries in elderly patients from 2001 to 2010 using a nationwide database: increasing incidence, overall mortality, and inpatient hospital charges. World Neurosurg. 2018, 120, e114–e130, doi:10.1016/j.wneu.2018.07.228.

To improve the manuscript, we have revised the sentences. (P4 L196–201, blue font)

This study required at least 1434 cases to achieve a medium effect (effect size, d = 0.25), a power of 0.90, and a statistical significance level of 0.05, considering the finding of a nationwide study that early mortality after traumatic cervical spine injuries in older patients was 11.3% [18]. In sample size calculation, we used the G-power software (Franz Faul, Univesitat kiel, Germany).

# Comment 2

A description of the missing data needs to be mentioned.

Response

We thank you for your important comment. As you mentioned, this study included cases with missing data due to the nature of this retrospective multicenter study. A pairwise deletion was performed in univariate analysis to address missing data. In multiple logistic regression analysis, listwise deletion of missing data was performed.

To improve the manuscript, we have modified the sentences. (P4 L191–192, blue font)

A pairwise deletion was performed in univariate analysis to address missing data. In multiple logistic regression analysis, listwise deletion of missing data was performed.

# Comment 3

The discussion section is too large; please reduce it. For example, there is no need to repeat the result part again.

Response

We thank you for your insightful comment. We deleted the part that repeats the fact already mentioned in the results section and the part deemed redundant.

To improve the manuscript, we have revised the sentences.

In this study, complete motor paralysis, chronic kidney disease, presence of cervical spine fracture, male sex, and older age were identified as risk factors for early mortality after traumatic cervical injury in older patients. (P8 L290–292, blue font)

This is the first study to explore the types of pre-existing comorbidities associated with early mortality after traumatic spine injury in older patients. Initially, we suspected that respiratory and cardiac diseases would be prognostic factors because early mortality related to respiratory and cardiac complications is common with traumatic cervical spine injuries. However, while the two comorbidities were significant in the univariate analysis, only , which identified chronic kidney disease was identified as an independent risk factor for early mortality in the multivariate analysis. (P8 L307–313, blue font)

In this subgroup analysis, intraoperative complications and cardiac disease were identified as independent predictors of early mortality in addition to complete motor paralysis and older age. (P10 L376–378, blue font)

# Comment 4
Language editing and spell check are required.

Response

We thank you for your insightful comment. We have performed the language editing and spell check. Please see the red font in revised manuscript.

We appreciate your reconsideration of our manuscript. We believe that this study’s topic will interest your readers and that the data are informative. We hope that the revised manuscript is suitable for publication in your journal.

Sincerely,